# Structure–Property Relationships for Fluorinated and Fluorine-Free Superhydrophobic Crack-Free Coatings

**DOI:** 10.3390/polym16070885

**Published:** 2024-03-24

**Authors:** Sevil Turkoglu, Jinde Zhang, Hanna Dodiuk, Samuel Kenig, Jo Ann Ratto Ross, Saurabh Ankush Karande, Yujie Wang, Nathalia Diaz Armas, Margaret Auerbach, Joey Mead

**Affiliations:** 1Plastics Engineering Department, University of Massachusetts Lowell, Lowell, MA 01854, USA; sevil_kaynar@student.uml.edu (S.T.); jinde_zhang@uml.edu (J.Z.); joann_rattoross@uml.edu (J.A.R.R.); saurabhankush_karande@student.uml.edu (S.A.K.); yujie.wang.new@gmail.com (Y.W.); nathalia_diazarmas@student.uml.edu (N.D.A.); 2Department of Polymer Materials Engineering, Shenkar College, Ramat Gan 5252626, Israel; hannad@shenkar.ac.il (H.D.); samkenig@shenkar.ac.il (S.K.); 3U.S. Army Combat Capabilities Development Command Soldier Center, Natick, MA 01760, USA; margaret.a.auerbach.civ@army.mil

**Keywords:** PFAS-free coatings, surface wetting, superhydrophobic coatings, crack-free coatings

## Abstract

In this study, particle loading, polyfluorinated alkyl silanes (PFAS or FAS) content, superhydrophobicity, and crack formation for nanocomposite coatings created by the spray coating process were investigated. The formulations comprised hydrophobic silica, epoxy resin, and fluorine-free or FAS constituents. The effect of FAS content and FAS-free compositions on the silica and epoxy coatings’ chemistry, topography, and wetting properties was also studied. All higher particle loadings (~30 wt.%) showed superhydrophobicity, while lower particle loading formulations did not show superhydrophobic behavior until 13% wt. FAS content. The improved water repellency of coatings with increased FAS (low particle loadings) was attributed to a combination of chemistry and topography as described by the Cassie state. X-ray photoelectron spectroscopy (XPS) spectra showed fluorine enrichment on the coating surface, which increases the intrinsic contact angle. However, increasing the wt.% of FAS in the final coating resulted in severe crack formation for higher particle loadings (~30 wt.%). The results show that fluorine-free and crack-free coatings exhibiting superhydrophobicity can be created.

## 1. Introduction

Superhydrophobic (SH) surfaces are known for enhanced water repellency with a water contact angle (WCA) larger than 150° and a sliding angle (SA) of less than 10° [1,2,3,4,5,6,7] and have attracted significant attention for various applications, such as self-cleaning [8,9,10], anti-corrosion [11,12], drag reduction [13,14], anti-biofouling [15,16], and anti-icing [17,18,19]. SH surfaces were first discovered on the lotus leaf [20,21]. Similar water-repellent behavior has also been seen on other biological surfaces, such as Cicada orni and Rhinotermitidae [22]. Low-surface-energy materials and the appropriate surface roughness are the two main factors determining the wettability of superhydrophobic surfaces [20,21,23]. By looking at nature, it is understood that a hierarchical surface composed of micro- and nano-level roughness is needed to create an interface configuration suitable for artificial SH surfaces [24,25,26,27,28]. Numerous efforts have been made to fabricate superhydrophobic surfaces with bottom-up approaches, including lithography [29,30,31], template-based techniques [32], plasma treatments [33], and top-down methods, such as layer-by-layer deposition [34], self-assembly [35], and nanocomposite coatings [36]. Among the techniques mentioned above, the nanocomposite coating approach is desirable, especially for industrial applications, because the method is straightforward and the relatively low cost makes it suitable for large-scale production when combined with conventional coating application methods, such as spray coating or dip-coating [14,19,37,38,39].

Generally, superhydrophobic nanocomposite coating compositions involve the use of various nanoparticles (NPs), such as silica, zinc oxide, titania, alumina, and indium tin oxide (ITO) [40]. Among these nanoparticles, silica nanoparticles are often preferred for fabricating superhydrophobic coatings due to their low cost, abundance, and ease of surface treatment [41]. The effect of particle loading on the wetting behavior of superhydrophobic nanocomposite coatings has been widely studied [42,43]. It was observed that increased particle loading increased the static water contact angle as a result of enhanced roughness [44]. The choice of binder used in the superhydrophobic coating is also important as it affects both the wetting behavior and the mechanical and chemical robustness of the final coating. Epoxy resin is often used as a binder for superhydrophobic coatings because of its desirable mechanical and bonding properties, thereby enhancing durability and robustness [45]. However, the high surface energy and high content of polar groups in epoxy resin results in smaller contact angles for epoxy-based coatings compared to those of coatings based on other low-surface-energy polymers.

The polyfluoroalkyl substances (PFAS or FAS) have been extensively used to prepare superhydrophobic coatings, as the fluorinated materials minimize water wetting due to fluorine’s extremely low surface energy (~10 mJ m^−2^) [46,47,48,49,50]. West et al. [51] reported a drastic increase in water contact angle (153°) with 2 wt.% fluoroalkyl silane treatment for a polyurethane-based coating that was intrinsically hydrophilic (60°) prior to the FAS treatment. Brassard et al. [52] prepared superhydrophobic coatings using functionalized fluorinated silica nanoparticles suspended in a solution and then deposited via spin-coating onto aluminum substrates. Their study showed that a critical roughness value of ~0.7 μm was required to achieve superhydrophobic behavior for their system. Researchers have reported the migration of the fluorinated chains to the surface in fluorinated systems [53,54]. Despite the advantages, there are various disadvantages to using fluorinated compounds in superhydrophobic coatings. For example, Fu et al. [54] presented superhydrophobic anti-icing coatings based on fluorinated polyurethane and hydrophobic fumed silica nanoparticles. Their study showed that increasing fluorine content decreased superhydrophobic stability and mechanical strength. They suggested that the decreased superhydrophobic stability and mechanical strength were the results of the low interfacial strength within the coating because of the addition of the low-surface-tension fluoroalkyl chains. High cost is another disadvantage of using fluorine materials in superhydrophobic surface fabrication [55,56]. Of major significance is that the perfluorinated chemicals used to tailor the surfaces are potentially harmful to the environment as long-chain perfluorinated alkyl substances are toxic and bioaccumulative [25,57,58]. This becomes particularly crucial in industries such as food production or textiles, where direct interaction with the human body is required. Therefore, there is an urgent need to investigate alternative superhydrophobic coatings that exhibit minimum toxicity [59].

To address the environmental concerns of fluorinated compounds and prepare environmentally friendly superhydrophobic coatings, non-fluorinated materials have been found in the literature [25]. Zhao et al. [60] prepared a fluorine-free polyethersulfone (PES)-based superhydrophobic coating using a spray coating method. Janowicz et al. [61] reported fluorine-free transparent superhydrophobic nanocomposite coatings from mesoporous silica (9 to 50 wt. %.) by spin coating and aerosol-assisted chemical vapor deposition (AACVD). Their study demonstrated that AACVD was preferred as superhydrophobicity was achieved at 9 wt. % particle loading for AACVD as compared to 41 wt. % for spin coating. Unfortunately, the chamber size limits the scale-up for large substrates. Zhao et al. [62] reported waterborne fluorine-free superhydrophobic coatings based on silica nanoparticles and silanes by spray coating. Wang et al. [63] prepared a superhydrophobic coating with silica nanoparticles and dimethyloctadecyl [3-(trimethoxysilyl) propyl] ammonium chloride to create multifunctional materials with mechanical, chemical, and physical robustness. In prior work, the behavior and quality of the coatings (superhydrophobicity and crack formation) as a function of the filler loading and FAS concentration have not been fully explored. The effect of FAS on the coating quality and topology will be particularly important to guide future industrial developments for crack-free coatings without the use of FAS materials. 

In this work, superhydrophobic coatings using non-hazardous solvents and fluorine-free (non-polyfluorinated alkyl substances—PFAS or FAS) and fluorine-containing formulations were compared by spray coating on glass substrates. The effect of FAS content on the chemistry, topography, wetting properties, and crack formation of the nanocomposite coatings comprised of hydrophobic silica and epoxy, with two different particle loadings, was investigated for comparison. Crack formation was also studied in the prepared coatings.

## 2. Experimental

### 2.1. Materials

Hydrophobic fumed silica nanoparticles (CAB-O-SIL TS-720) with a particle size of 22 nm were purchased from Cabot Corporation, Billerica, MA, USA. Isopropanol (IPA) (ACS reagent, 99.5%) was obtained from Sigma Aldrich (St. Louis, MO, USA). A two-component epoxy (EPO-TEK 301) consisting of part A (Bisphenol-A diglycidyl ether) and part B (Triethyl-1,6 hexanediamine) was purchased from Epoxy Technology Inc. (Billerica, MA, USA).The Fluoroalkylsilane (FAS)/isopropanol solution, specifically Dynasylan^®^ F8263, Triethoxy(3,3,4,4,5,5,6,6,7,7,8,8,8 tridecafluorooctyl) silane, was purchased from Evonik Inc. (Parsippany, NJ, USA). Plain glass microscope slides (75 × 25 mm) were used as substrates (Fisher Scientific Company, Hampton, NH, USA, Cat. No. 12-550-A3).

### 2.2. Preparation of Coatings

The non-fluorinated silica/epoxy coating was prepared by premixing EPO-TEK 301-part A and part B in a 4:1 ratio at 20 °C. The mixture was stirred for 10 min at a speed of 450 rpm at a temperature of 20 °C. Subsequently, hydrophobic fumed silica NPs (CAB-O-SIL TS-720, Cabot Inc., Billerica, MA, USA), isopropanol (ACS reagent, ≥99.5%, Sigma-Aldrich), and the blended epoxy were stirred together in a beaker for 60 min to improve the homogeneity of the suspension. Isopropanol, a widely used solvent with excellent solubility properties for both the binder and particle, was incorporated into the mixture to facilitate the dispersion of silica nanoparticles and ensure uniform blending with the epoxy matrix. During this mixing step, fluoroalkylsilane (FAS)/isopropanol solution was introduced into the suspension mixture for the fluorinated coatings. Sonication, a powerful technique based on the application of high-frequency sound waves, was employed to further disperse and deagglomerate nanoparticles within the suspension. The suspension was sonicated for 5 min at a frequency of 40 kHz and an amplitude of 50%. A high-volume, low-pressure (HVLP) spray gun (DeVilbiss 802342 Starting Line HVLP Gravity Spray Gun, DeVilbiss, Somerset, PA, USA) was used to coat the suspension on the glass slides. Before coating, glass substrates were cleaned with isopropanol and dried with pressurized air. The spray parameters, including the spray pressure, the distance between the spray gun nozzle and substrate, and spray speed, were carefully optimized to achieve uniform coverage and thickness across the substrate surface. For spray coating, the spray pressure was set at 207 kPa (30 psi). The distance between the spay gun nozzle and substrate was 20 cm, and the spray speed was 76 mm/s (3 in/s). Samples were sprayed once and then dried at 110 °C for 2 h to ensure complete epoxy curing in the final coating. The compositions of the prepared coatings are presented in Table 1. 

The reason behind selecting particle loadings of ~15% and ~30% in this paper is rooted in the prior literature, which indicated that a turning point for achieving superhydrophobicity occurs around 25% particle loading [64]. In addition, while most studies have employed FAS content within the range of 2–5%, this work intentionally chooses a higher FAS content to investigate its impact on wetting behavior [52].

### 2.3. Characterization

The formulated coatings were characterized using several techniques. The water contact angle (WCA) measurements were conducted using the sessile drop method with a Drop Shape Analyzer (DSA100-KRÜSS GmbH, Hamburg, Germany). At a temperature of 20 °C for all coatings to characterize the surface properties. The temperature was rigorously controlled at 20 °C to eliminate any thermal effects on surface properties. The static WCA measurements were performed with a 5 µL droplet volume, while the sliding angle (SA) measurements were taken with a 20 µL droplet volume. Multiple measurements were taken across different regions of each coated substrate to capture spatial variations in surface wettability.

Scanning electron microscope (SEM) images were taken on a field-emission scanning electron microscope (JSM 7401F, JEOL Inc. Peabody, MA, USA), typically at an electron energy from 2 to 10 kV, to examine the morphology of the coating surface. The electron dispersive spectroscopy (EDS) analysis was performed to obtain the elemental composition of the coatings. The EDS analysis was performed concurrently with the SEM imaging to elucidate the elemental composition of the coatings. The technique enabled qualitative and quantitative assessments of elemental distribution and concentration within the coating matrix.

The topography of the coating surface was analyzed using a confocal laser microscope (CLM) (LEXT 243 OLS5000, Olympus Inc., Center Valley, PA, USA). In order to provide a thorough evaluation of the surface features, roughness measurements were carried out by scanning a 259 × 259 μm^2^ area. Furthermore, to ensure the reliability and accuracy of the results, roughness parameters were obtained by scanning at least three samples for each formulation, thereby enhancing the robustness of the findings.

An X-ray photoelectron spectroscopy (XPS) was carried out using a Sigma Probe Thermo V.G. Scientific instrument. The base pressures in the analysis and preparation chambers were approximately 10^−10^ and 10^−9^ mbar, respectively. Al Kα X-rays were employed with a pass energy of 100 eV and 20 eV for the survey and elemental analysis, respectively. The samples were scanned for binding energies ranging from 0 to 900 eV. The survey and elemental analysis were carried out with 0.5 eV and 0.05 eV energy step sizes, respectively, with a dwell time of 50 ms. The number of scans (4–50) was based on the concentration of elements present in the sample. The curve fitting and data analysis were performed using Avantage V6 software. The surface charge correction was performed by aligning the C1s peak of aliphatic carbon to 284.6 eV. 

## 3. Results and Discussion

The wetting behavior (water contact angle and sliding angle) for the silica/epoxy nanocomposite coatings is shown in Table 2. The results in Table 2 show that all coatings with a higher particle loading (30 wt. %) exhibit superhydrophobic behavior, regardless of FAS content. In contrast, low particle loading samples (15 wt. %) show superhydrophobic behavior only at levels of 13 wt. % FAS. One key finding from Table 2 is that achieving superhydrophobicity without FAS is possible, but it requires a higher particle loading (H-F0). Conversely, lower loadings are sufficient when FAS is present. Both FAS and particle loading impacted wetting, with particle loading playing a more dominant role.

The measured apparent contact angle increases with an increase in FAS wt.% in the formulation. The increase in FAS content in the coating formulation decreases the intrinsic surface energy, leading to a higher intrinsic angle in the final dry coating, according to Young’s equation (Equation (1)) where $\gamma_{SV}$, $\gamma_{SL}$, and γ are the solid–vapor, solid–liquid, and liquid–vapor interfacial surface tensions, respectively. Ultimately, a higher intrinsic angle leads to an increased apparent contact angle, as described by the Wenzel and Cassie equations (Equations (2) and (3), respectively) where $\theta^{*}$ and θ represent the apparent and intrinsic (Young’s) contact angles, respectively; r is the roughness factor, which represents the ratio of the actual surface area over the projected surface area; and $\phi_{s}$ is the solid–liquid interface fraction [65]. In the case of superhydrophobic surfaces, the Cassie equation is the dominant mechanism to consider (Equation (3)) [42].
(1)$$\cos\theta =\frac{\gamma_{SV}-\gamma_{SL}}{\gamma }$$
(2)$$cos\theta^{*}=r\;cos\theta$$
(3)$${cos}^{*}=\phi_{s}\cos\theta +\left(1-\phi_{s}\right)$$

To investigate whether the fluorine materials were present on the surface, and, thus, lowering the surface energy of the coatings, we used XPS to evaluate the atomic percentages on the surface. The Al Kα XPS Survey figures of 2 wt.%, 5 wt.%, 9 wt.%, and 13 wt.% FAS (Figure 1a–d) confirm the presence of fluorine, silicon, carbon, oxygen, and nitrogen in the coatings. In addition to the wide energy range survey spectra, the high-energy resolution spectra of the characteristic peaks of elements such as C 1s, O 1s, N 1s, F 1s, and Si 2p were recorded through a narrow energy range. The Al Kα Si 2p spectrum of 2 wt.% FAS shows two peaks at 102.1 eV and 103.5 eV, as seen in Figure 1. The peak at 103.5 eV corresponds to a typical peak of silica, while the peak at 102.1 eV represents the silanes from FAS. The Al Kα F 1s spectrum of 2 wt.% FAS shows two peaks at 689.9 eV and 688.1 eV, corresponding to fluorine from –CF_2_ and –CF_3_ elements of FAS. The Si 2p and F 1s peaks observed for 2 wt.% FAS are also seen in the spectrum of 5 wt.%, 9 wt.%, and 13 wt.% FAS at a similar binding energy.

Figure 1 also summarizes the atomic percentages of several elements. It shows that %F on the coating surface increases as the %FAS increases in the formulation (from 8% to 20%). Additionally, for all samples, the reported %F in XPS data was much larger than the calculated value from the formulation. This shows that the fluoroalkyl silane has migrated to the surface and enriched the fluorine content of the coating surface, ultimately leading to a lower energy surface and higher contact angle based on the material alone, without considering the surface roughness. 

XPS captures the percentage of fluorine on the surface (Figure 1), which specifically examined a depth of approximately 10 nanometers [66]. To examine the fluorine distribution through the thickness, EDS analysis was performed on the upper, middle, and bottom regions of the cross-section (Figure 2) of the H-F13 coating. Looking at locations below the surface (considered the bulk), we see that the variation in fluorine atomic percentages only vary in the second valid digit and are approximately 1% across different regions of the coating, namely the upper, middle, and bottom regions. We consider that the fluorine is relatively evenly distributed throughout the bulk of the coating (disregarding the surface). 

The atomic percentages (At. %) of the elements are also presented in Figure 2. The data showed a small At. % of fluorine (~2%) that was smaller than the calculated fluorine At. % value (~5%) based on the content of the fluorinated species in the formulation. These data support the conclusion that the FAS has preferentially migrated to the surface of the coating, thereby enriching the fluorinated species on the surface and resulting in lower surface energy. 

While the presence of fluorine changes the inherent surface tension of the material, the roughness of the surface also affects the superhydrophobicity. Traditionally, there are two different wetting models, the Wenzel model and the Cassie–Baxter model, used to describe the interaction of liquid drops with a hydrophobic surface and predict the equilibrium water contact angle on surfaces [67]. According to the Wenzel model, a liquid drop completely penetrates the roughness and is strongly pinned by the roughness. The Cassie–Baxter (C–B) state is preferred for superhydrophobic surfaces, as it has a trapped air layer (air plastron or air cushion) and provides better water repellency. Air trapped underneath the liquid drop provides a small contact angle hysteresis in addition to a large contact angle. 

The topography of the prepared coatings was analyzed, and their surface statistical parameters were determined using Olympus software (OLS5000 Ver.1.3.5). The investigation focused on examining the relationship between the topography and the water repellency of the coatings. Three-dimensional images of the coating surface obtained via confocal laser microscopy are illustrated in Appendix A. Various roughness parameters, including the roughness factor (r), the root-mean-square values of roughness (Sq), and the autocorrelation length (Sal), were extracted and their relationships with particle loading and FAS content are illustrated in Appendix A, Appendix A, and Figure 3, respectively.

Among these roughness parameters, Sal represents the distance between surface asperities and is commonly used to correlate with the wettability of a surface. It serves as a crucial surface parameter, with a significant impact on the water repellency of the surface. It has been previously shown that surfaces with a smaller Sal could have larger breakthrough pressure, thereby increasing robustness [64].

Figure 3 illustrates that higher particle loading coating results in a smaller Sal (9 µm). A small Sal refers to a smaller distance between the peaks, resulting in a higher breakthrough pressure and a more stable Cassie–Baxter state [68]. This suggests that higher particle loadings provide better robustness and water repellency. 

The results show that the addition of fluorine has a limited effect on the Sal values. The Sal values are dominated by the particle loading, which in turn affects superhydrophobicity. The data also show that despite the relatively large Sal (18 µm) values for the low particle loading, the addition of fluorinated species reduces the surface energy sufficiently to result in the Cassie state and superhydrophobicity. 

Since the quality of the coatings is important for applications, the morphology of the prepared silica/epoxy nanocomposite coatings was characterized by FE-SEM, and the results are illustrated in Figure 4 and Figure 5. It was observed that all coatings had a hierarchical structure with micro- and nanoscale roughness, as depicted in Figure 4 and Figure 5. In the higher loadings, the nanoparticles are more closely packed, which is supported by the lower Sal values. This hierarchical structure was found to provide better water repellency. Furthermore, coatings with higher particle loading (~30 wt.%) had more silica nanoparticles on the surface than those with lower particle loading (~15 wt.%). 

Similar to particle loadings, FAS content also had an impact on morphology. In Figure 4, we see that for the low particle loadings (15 wt.%), the coatings show no obvious cracks, even at high fluorine contents (up to 13 wt.% FAS). On the other hand, for high particle loadings (30 wt.%) (Figure 5), we begin to see the presence of cracks beginning at low fluorine contents (2 wt.% FAS) and becoming larger and more prevalent as the FAS content increases. While FAS enhances water repellency, using higher concentrations raises environmental concerns and can lead to cracks in the coating. These cracks are detrimental to performance, especially in situations involving ice and corrosion [69,70].

Cracks in the coating can be attributed to two factors: the particle loading and the presence of FAS. Cracks primarily result from high particle loading, where insufficient binder leads to a weakened coating. Elevated particle contents induce stress, ultimately leading to crack formation [70]. To better understand the reason behind the crack formation, the cross-section of the coatings was analyzed using SEM. The cross-section images of the coatings are presented in Figure 6. Figure 6 shows visible cracks along the thickness direction, which would allow for the passage of molecules, such as water, to reach the substrate below and reduce the coating’s protective effectiveness (e.g., corrosion). Hence, the crack formation mechanism could be attributed to the reduced interfacial adhesion due to the presence of fluorinated species affecting the bonding between the epoxy resin and hydrophobic silica nanoparticles. 

Both a higher particle loading and an increased FAS content contribute to the formation of cracks. Managing these factors is crucial to prevent cracks and ensure the durability and performance of the coatings.

## 4. Conclusions

In this study, particle loading, FAS content, superhydrophobicity, and crack formation for nanocomposite coatings created by the spray coating process were studied. Hydrophobic silica particle loading affected both wetting and topography. All higher particle loadings (~30 wt.%) showed superhydrophobicity, while lower particle loading formulations did not show superhydrophobic behavior until 13% wt. FAS content. Compared to the lower particle loading systems, higher particle loading formulations showed a smaller autocorrelation length (Sal) that could enhance the robustness and, thus, superhydrophobicity of the coatings. It was found that the addition of fluorine had a limited effect on the Sal values and that the Sal values were determined by the particle loading. Since the Cassie state (required for superhydrophobicity) is dependent on the chemistry and topography of the surface, the improved water repellency of coatings with increased FAS (low particle loadings), despite the relatively large Sal (18 µm) values for the low particle loading, shows that the addition of fluorinated species reduces the surface energy sufficiently to result in the Cassie state and superhydrophobicity. 

XPS spectra showed a fluorine enrichment on the coating surface, which increases the intrinsic contact angle. However, increasing the wt.% FAS in the final coating resulted in severe crack formation for higher particle loadings (~30 wt.%). Cracks in the coating were attributed to both the particle loading and the presence of FAS at the particle interface. The results show that fluorine-free and crack-free coatings exhibiting superhydrophobicity can be created.

## Figures and Tables

**Figure 1 polymers-16-00885-f001:**
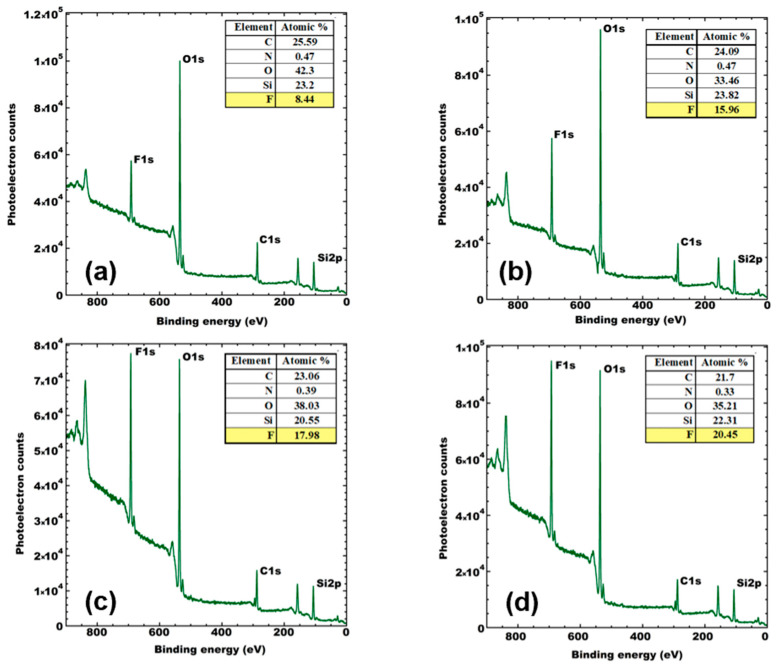
X-ray photoelectron spectroscopy (XPS) survey spectra of the samples (**a**) H-F2, (**b**) H-F5, (**c**) H-F9, and (**d**) H-F13.

**Figure 2 polymers-16-00885-f002:**
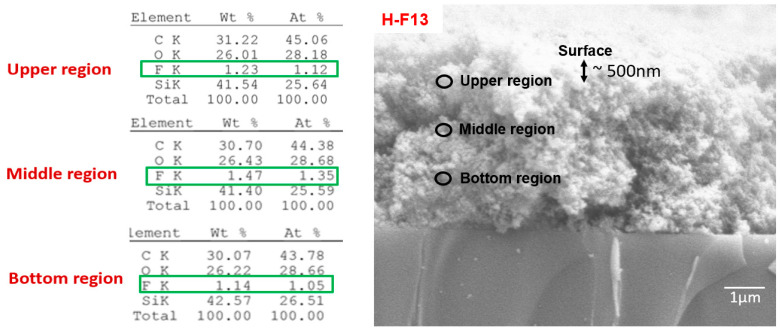
Electron dispersive spectroscopy (EDS) analysis of the cross-section of H-F13. The left image depicts elemental analysis, while the right image shows the cross-section of H-F13.

**Figure 3 polymers-16-00885-f003:**
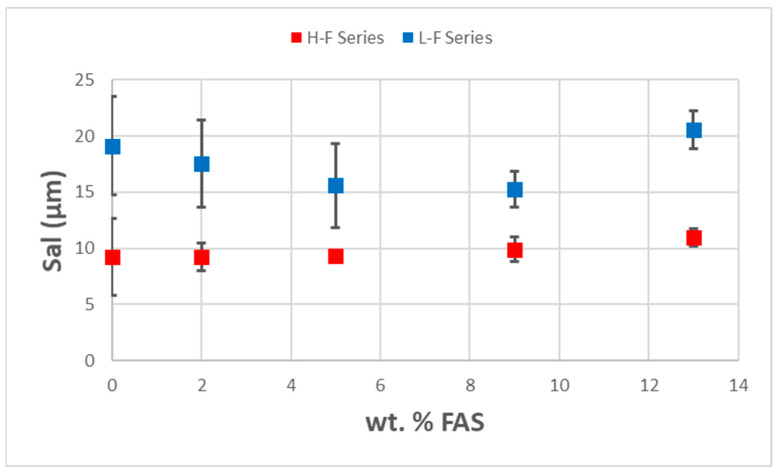
The relationship of Sal with FAS content and particle loading.

**Figure 4 polymers-16-00885-f004:**
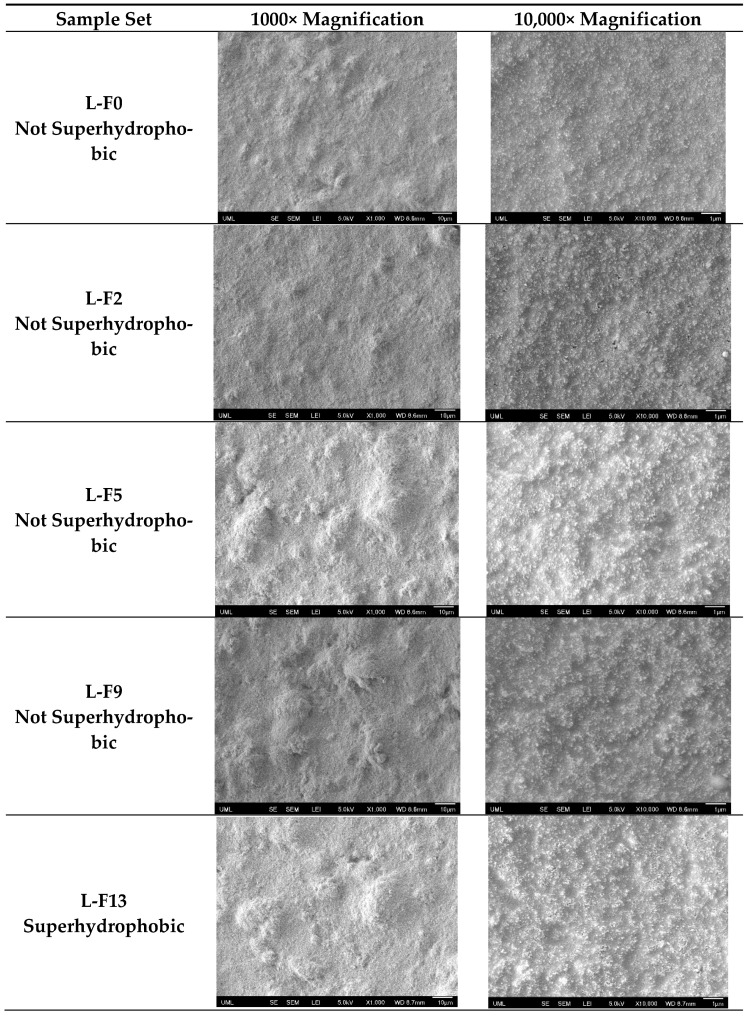
Scanning electron microscopy (SEM) top-view images of several samples in the L-F seriesHigher resolution image provide. The scale bar represents 10 microns for the left images and 1 micron for the right images.

**Figure 5 polymers-16-00885-f005:**
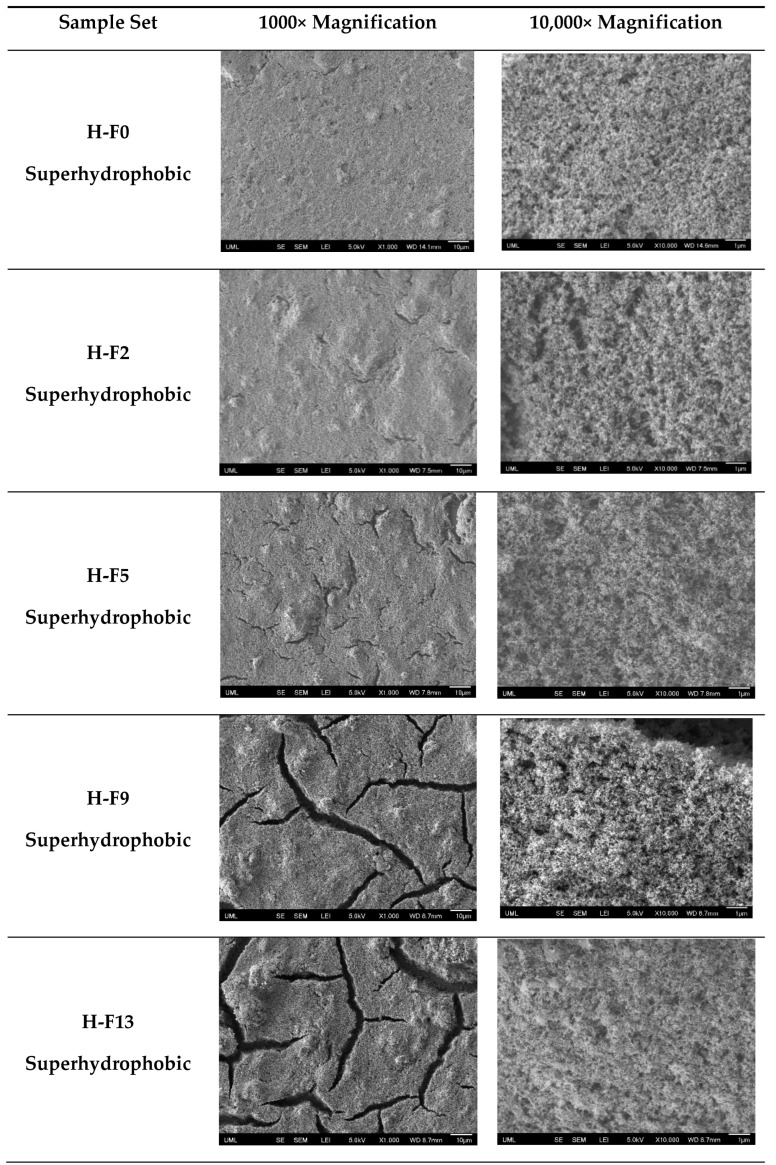
SEM top-view images of several samples in the H-F series. The scale bar represents 10 microns for the left images and 1 micron for the right images.

**Figure 6 polymers-16-00885-f006:**
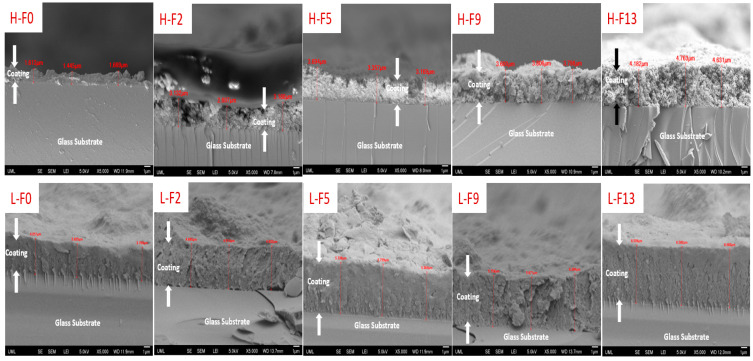
SEM cross-section images of the coatings. The scale bar represents 1 micron for all images.

**Table 1 polymers-16-00885-t001:** Compositions of Coatings.

Sample	Silica (g)	Epoxy (g)	Fluoroalkylsilane (FAS)/Isopropyl Alcohol (IPA) Solution (4 wt.%) (mL)	IPA (mL)	Particle to Binder (P:B) Ratio	FAS (wt.%)
L-F0	0.90	6	0	70	(3:17) ~15 wt.% silica	0
L-F2	0.90	6	3.10	66.9		2
L-F5	0.90	6	7.75	62.25		5
L-F9	0.90	6	15.50	54.5		9
L-F13	0.90	6	23.25	46.75		13
H-F0	1.89	4.11	0	110	(3:7) ~30 wt.% silica	0
H-F2	1.89	4.11	3.10	106.9		2
H-F5	1.89	4.11	7.75	102.2		5
H-F9	1.89	4.11	15.50	94.5		9
H-F13	1.89	4.11	23.25	86.7		13

**Table 2 polymers-16-00885-t002:** The effect of coating composition on wetting behavior.

Sample Set	Contact Angle (°)	Sliding Angle (°)	Superhydrophobicity
L-F0	128 ± 2	>60	No
L-F2	143 ± 2	>60	No
L-F5	158 ± 1	>60	No
L-F9	161 ± 2	>20	No
L-F13	163 ± 2	<5	Yes
H-F0	161 ± 4	<5	Yes
H-F2	163 ± 2	<5	Yes
H-F5	164 ± 3	<5	Yes
H-F9	163 ± 2	<5	Yes
H-F13	165 ± 3	<5	Yes

## Data Availability

Data are contained within the article and Appendix A.

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
