# Peer review of "Structure–Property Relationships for Fluorinated and Fluorine-Free Superhydrophobic Crack-Free Coatings"

_polymers, 2024, doi:10.3390/polym16070885_

Round 1

Reviewer 1 Report

Comments and Suggestions for Authors

(1) How can the FAS ratio in table 1 be calculated? I attempted a simple calculation, but I was unable to obtain the numbers listed. (2) The sentences seem to be contradictory. One sentence states, "This showed that the fluorine is evenly distributed in the bulk coating," while the other sentence suggests, "the FAS has preferentially migrated to the surface of the coating" (3) The paper does indeed present a coherent story; however, it lacks sufficient details. Additional characterizations, such as those related to the adhesion and durability of the coating, should be included

Reviewer 2 Report

Comments and Suggestions for Authors

The manuscript by Mead et al. investigated superhydrophobicity, crack formation for nanocomposite coatings created by the spray coating process from a variety of FAS-loaded composite. The methodologies used are well but not sufficient. The works seems acceptable after major revision.

The molecular structure and composite formation scheme must be presented.

FTIR spectra must be recorded at least for two extreme wt% cases to characterize the characteristic stretching bands of the precursor and composites.

Do the films show stability against UV radiation. One recording of wetting experiment could be performed to prove that fluorinate and non-fluorinated coating are air+UV stable.

A plenty of decade old citations have been added while the manuscript is missing important citations like:

Langmuir 2023, 39, 22, 7731–7740

Sci Rep 12, 17059 (2022) etc.

Therefore, I suggest major revision.

Comments on the Quality of English Language

The current writing is very formal, it should be edited with a plenty of attentions.

Reviewer 3 Report

Comments and Suggestions for Authors

The article is relatively short (14 pages, without references 11 pages, a third of them are images).

The introduction is documented by an appreciable number of journal references, no book reference is included. In my opinion, including also books representing basic related literature or monographies would be beneficial, however 4 references have been identified as review articles.

What is the core of the article is described in the last paragraph of Introduction: A comparison of selected superhydrophobic coatings by spray coating on glass substrates. If I understand well, they do not present new methods, but prepare and compare materials using methods referred in the previous part.

Used materials are described sufficiently for the purpose, however if authors can report on size, shape and distribution of nanoparticles in section 2.1, and specify components of EPO-TEK 301, the value of the article would increase.

In section 2.2, the temperature of mixing and stirring coating components should be specified, although laboratory temperature is implicitly understood. Stirring intensity could be somehow specified, since one can imagine very different speeds, when the resulting properties may by different. Similarly, when authors write "suspension sonicated for 5 min", they could add more parameters than duration.

For contact angle measurement, I am lacking measurement temperature either in section 2.3 or in result presentation.

If authors present K-Ratio in Figure 2, they should write one short sentence explaining its meaning and role; if they state (lines 202-203) that "This showed that the fluorine is evenly distributed in the bulk coating.", they should first sum up that contents differs in second valid digit, which the authors consider to be evenly distributed. The similar is valid for other conclusions made from the raw data; e.g. for figures 4 and 5, they should describe the elements and phenomena proving what the figure documents instead of direct stating conclusions based on figure contents.

Since the authors refer to Figure S5 in the main text, it is uncomfortable to look for it in a separate file - all what is directly mentioned in the main text, should be either in the main text, or appended after references in the same file to be easily accessible to the reader.

In the Conclusions section, the sentence "The improved water repellency of coatings with increased FAS (low particle loadings) was attributed to a combination of chemistry and topography as described by the Cassie state." should be complemented by more detailed elaboration probably of lines 228-236 to make more straightforward the connection between this conclusion and results.

Formal issues:

The author instructions state:"Acronyms/Abbreviations/Initialisms should be defined the first time they appear in each of three sections: the abstract; the main text; the first figure or table. When defined for the first time, the acronym/abbreviation/initialism should be added in parentheses after the written-out form."
No exception is specified even for well known ones.
I do not see where it is done for IPA, and in the first figure or table in this article. XPS is in full in the main text only, not in abstract, EDS is not stated in full, neither At. I suggest to check again all abbreviations, in particular in tables and figures, whether they are defined not only in the main text, but also at the first figure or table where they are used.

In writing equations and quantities, IUPAP and IUPAC recommendations are not fully observed.
The IUPAP recommendation is available for example at https://iupap.org/wp-content/uploads/2021/03/A4.pdf.
The IUPAC recommended documents are Quantities, Units and Symbols in Physical Chemistry (https://iupac.org/what-we-do/books/greenbook/)
and On the use of italic and roman fonts for symbols in scientific text ( https://iupac.org/wp-content/uploads/2016/01/ICTNS-On-the-use-of-italic-and-roman-fonts-for-symbols-in-scientific-text.pdf)

In lines 166 and 172 in eq. (1)-(3), CV, SL, S subscripts should be upright, cos also upright since it denotes a function. Similarly quantities in lines 217-219; for physical quantities, standard symbols should be preferentially used instead of their own (exactly - taken from other authors not following standards) abbreviations (e.g. for autocorrelation length L for length with subscript denoting that it is autocorrelation length).

In Figure 1, text elements are too small and therefore not optimally readable.

To conclude:

The article is interesting and bringing useful knowledge, however the authors should try to specify more exactly used materials and methods and to describe the way from raw results to final conclusions in more details. Fixing formal imperfections will be welcome.

Reviewer 4 Report

Comments and Suggestions for Authors

In this work, superhydrophobic coatings using non-hazardous solvents and fluorine-free (non-polyfluorinated alkyl substances - PFAS or FAS) and fluorine-containing formulations were compared by spray coating on glass substrates. The author makes a detailed analysis. However, the reviewer does not find the innovation contents and specific application scenarios of this article. It is a common technique to prepare the coating by mixing the existing superhydrophobic micro-nanomaterials. Therefore, the author needs to figure out the starting point of the article, or combine some new processes, such as additive manufacturing and 4D printing, to prepare innovative coating layers. (https://doi.org/10.1016/j.addma.2020.101615, https://doi.org/10.1016/j.jmst.2022.03.009, https://doi.org/10.1002/adma.202200750) These articles help to cross-discipline, combining additive manufacturing with three-dimensional coatings.

There are some errors in the writing specification. The scale bars in the SEM images Fig. 4 and Fig. 5 are not standard.

The annotation in Figure 2 lacks specification. The left image is elemental analysis and the right image is not annotated. Also, the scale bar in the right image is contradictory to the surface's 300 nm.

Round 2

Reviewer 1 Report

Comments and Suggestions for Authors

Accept this submission as it is

Author Response

We express our gratitude to the reviewer for their valuable contributions to our manuscript.

Reviewer 2 Report

Comments and Suggestions for Authors

The manuscript has the required documents and clarity now. Conclusion has also been improved and more specific to this work. I suggest accept.

Author Response

(The authors gave the same response as above.)

Reviewer 3 Report

Comments and Suggestions for Authors

The authors have mostly changed their text according to the comments or explained their choice.
However, the following comment has been omitted:

If the authors state (lines 209-210) that "This showed that the fluorine is evenly distributed in the bulk coating.", they should first sum up that contents differs in second valid digit. Only after this, they should  write that they consider to be evenly distributed. Similarly in other conclusions made from the raw data; e.g. for figures 4 and 5, they should describe the elements and phenomena proving what the figure documents instead of direct stating conclusions based on figure contents.
(lines changed according to the new version)

Only tiny formal comments remain:
Some pieces of added text are of different size than other text, the size should be unified before publication.

Value and unit should be divided by a space in lines 108 (nm), 109 (%), 117-118 (°C), 123 (%), 128 (°C, h), 131-134 (%), 140 (°C), 145 (kV) and many other.  Please, check through the whole text.

According to journal requirements, following abbreviations should be again defined at the first usage in table or figure, although they are already defined in the main text and some of them are well known:
Table 1: FAS and IPA
Figure 1: XPS
Figure 2: EDS
Figure 4: SEM

In Figure 1, yet larger text elements would be optimal in my opinion.
In Figures 4-6, the bottom bars of the photographs with their data are not comfortably readable

In Ref. 2, RSC instead of Rsc should be in my opinion in the abbreviated journal name.
The authors can consider unification of capital letters writing in the journal names and article titles.

Reviewer 4 Report

Comments and Suggestions for Authors

The article has been revised according to the comments.

Author Response

(The authors gave the same response as above.)
